# Four eyes see more than two: Dataset Distillation with Mixture-of-Experts

## Abstract

The ever-growing size of datasets in deep learning presents a significant challenge in terms of training efficiency and computational cost. Dataset distillation (DD) has emerged as a promising approach to address this challenge by generating compact synthetic datasets that retain the essential information of the original data. However, existing DD methods often suffer from performance degradation when transferring distilled datasets across different network architectures (i.e. the model utilizing distilled dataset for further training is different from the one used in dataset distillation). To overcome this limitation, we propose a novel mixture-of-experts framework for dataset distillation. Our goal focuses on promoting diversity within the distilled dataset by distributing the distillation tasks to multiple expert models. Each expert specializes in distilling a distinct subset of the dataset, encouraging them to capture different aspects of the original data distribution. To further enhance diversity, we introduce a distance correlation minimization strategy to encourage the experts to learn distinct representations. Moreover, during the testing stage (where the distilled dataset is used for training a new model), the mixup-based fusion strategy is applied to better leverage the complementary information captured by each expert. Through extensive experiments, we demonstrate that our framework effectively mitigates the issue of cross-architecture performance degradation in dataset distillation, particularly in low-data regimes, leading to more efficient and versatile deep learning models while being trained upon the distilled dataset.

## 1 Introduction

While large datasets like ImageNet (Deng et al., 2009) have demonstrably fueled the success of deep learning models, their sheer size presents a significant challenge in terms of training efficiency. The computational resources required to train models on these massive datasets can be prohibitively expensive, both in terms of time and financial cost. This burden is further amplified in scenarios like continual learning, hyperparameter optimization, and architecture search, where iterative training processes are essential. As the demand for data in deep learning continues to escalate, finding innovative solutions to mitigate the associated training costs becomes increasingly critical.

Two primary approaches have emerged to tackle the challenge associated with large datasets in deep learning: dataset distillation (DD) (Wang et al., 2018; Zhao & Bilen, 2021b; 2023; Cazenavette et al., 2022; Kim et al., 2022; Zhou et al., 2022) and coreset selection (Coleman et al., 2020). Coreset selection focuses on identifying a highly representative subset of the original data, known as a coreset. However, due to storage constraints, this subset may only capture a limited amount of information, potentially failing to well represent the complexities of the entire dataset. DD offers an alternative solution by generating a distilled synthetic dataset optimized to preserve model performance (i.e. the model trained upon original dataset and the one trained upon distilled dataset ideally should perform similarly). While both DD and coreset selection operate under the same storage limitations, DD has the advantage of encoding a richer set of information and details compared to a coreset, thus benefiting the downstream tasks such as image classification, continual learning (Yang et al., 2023; Wiewel & Yang, 2021), and federated learning (Zhang et al., 2022; Xiong et al., 2022).

Since its inception by (Wang et al., 2018), dataset distillation has witnessed a surge in research efforts, with various approaches exploring different facets of the technique. These approaches can be

broadly categorized into three main streams: **1) Gradient Matching** which basically aims to align the gradients generated from synthetic and real data for guiding the distillation process (Zhao & Bilen, 2021b; Kim et al., 2022); **2) Trajectory Matching** which attempts to align the training trajectories of synthetic and real datasets, ensuring that the optimization path of the model learnt from the distilled data closely resembles that of the original data (Cazenavette et al., 2022); **3) Distribution Matching** which focuses on matching the latent distribution of features extracted from real and synthetic data, encouraging the distilled data to capture the underlying statistical properties of the original data. These diverse approaches highlight the ongoing efforts to refine and optimize DD techniques, facilitating the generation of more compact and informative synthetic training datasets to fuel the deep model training.

While the conventional wisdom in deep learning suggests that larger models generally outperform smaller models when being trained on a given dataset, dataset distillation (DD) presents a contrasting scenario. Synthetic datasets generated through DD often exhibit a tendency to overfit to the specific neural network architecture used during the distillation process. This phenomenon explains the prevalent use of small architectures like ConvNet-3 in DD research for performance comparison. However, when these distilled datasets are transferred to larger and more complex architectures, such as ResNet-18 (He et al., 2016), AlexNet (Krizhevsky et al., 2012), or VGG-11 (Simonyan & Zisserman, 2015), a significant degradation in classification performance is often observed. This limitation poses a significant challenge to the broader applicability of DD, as it restricts the transferability of distilled knowledge across diverse model architectures. Consequently, there is a pressing need for novel DD methods that explicitly address this issue and promote generalizability across a wider range of network architectures. To address this challenge, we propose a novel mixture-of-experts (MoE) framework for dataset distillation, drawing inspiration from (Zhen et al., 2022) and (Zhou et al., 2020). Our approach leverages the concept of dividing the distillation task among multiple expert models, each focusing on a distinct subset of the original data. This division of labor encourages specialization within each expert, potentially capturing different aspects of the data distribution and leading to a more diverse and informative distilled dataset, in which the enhanced diversity results to benefit the cross-architecture generalizability. For instance, under the constraint upon Image Per Classes (IPC), we split the distillation process into two parallel paths (two experts), with having each expert model responsible for distilling the original dataset into half of the allocated storage budget (noting that overall storage budget, i.e. the number of distilled images, hence is kept the same as the original dataset distillation method which does not have multiple experts). To further promote diversity within the synthetic dataset, we introduce a distance correlation minimization strategy: during each iteration of the distillation process, the distance correlation between the expert models is calculated and subsequently minimized. This step encourages the experts to generate synthetic data that is dissimilar across experts, thereby enriching the overall information content captured by the entire distilled dataset.

Finally, while one attempts to leverage the distilled dataset to train a new model, the fusion strategy based on Mixup (Zhang et al., 2018) can be seamlessly introduced to combine the synthetic images from each expert, thanks to the complementary information captured across individual experts, leading to a more comprehensive and generalizable representation of the original data. To evaluate the effectiveness of our proposed framework, we conducted experiments with various IPC settings as well as adopting different dataset distillation methods to be experts, comparing the performance of single-expert and multiple-expert distillations. Notably, we observed a significant improvement in average cross-architecture performance with multiple experts, particularly in lower IPC conditions (e.g. $\leq 25$). This finding suggests that our approach effectively mitigates the transferability issue by promoting diversity and generalizability within the distilled dataset.

In summary, our work makes the following key contributions:

- We introduce a novel mixture-of-experts framework for dataset distillation that addresses the challenge of cross-architecture performance degradation.

- Our approach leverages distance correlation minimization and a fusion strategy to promote diversity and enhance the generalizability of the distilled dataset.

- We demonstrate the effectiveness of our framework by evaluating its performance with popular surrogate distillation methods, including MTT (Cazenavette et al., 2022), IDC (Kim et al., 2022), and IDM (Zhao et al., 2023).

## 2 RELATED WORK

**Dataset Distillation.** Pioneered by Wang et al. (2018), dataset distillation (DD) is a method that aims to condense large training datasets into a compact and informative synthetic set, enabling models trained on this distilled data to achieve comparable performance to those trained on the full dataset. The core of DD lies in a bilevel optimization framework. The inner loop minimizes the training loss of a model using distilled data, while the outer loop optimizes the distilled data based on the matching objective. Several studies have explored various objectives to enhance DD performance: **1) Gradient Matching** – DC (Zhao & Bilen, 2021b) optimizes synthetic images by matching gradients of neural networks' weights that are trained on real and distilled data, while DSA (Zhao & Bilen, 2021a) incorporating differentiable data augmentation to synthesize more informative images. IDC (Kim et al., 2022) follows DC but particularly starts the iterative procedure of dataset distillation via training the networks with real data (instead of the synthetic ones) to ensure the quality of gradients. Moreover, under limited storage constraints, IDC stores synthetic images in a low-resolution format, which will be up-sampled before being used to train a new network; **2) Trajectory Matching** – MTT (Cazenavette et al., 2022) starts from training a set of classification models on real data, recording their optimization trajectories. Subsequently, the distillation process focuses on aligning the training trajectories of models trained on the distilled data with those obtained from real data. Due to the reduced size of the distilled dataset, the training steps for synthetic data are typically larger (approximately 10x to 30x) than those used for real data. FTD (Du et al., 2023) further refines MTT by addressing its issue of having interrupted trajectories between iterations, promoting a smoother and flatter trajectory for enhanced performance; **3) Distribution Matching** – DM (Zhao & Bilen, 2023) and IDM (Zhao et al., 2023) aim to align the feature representations learned from real and synthetic datasets, while IDM enhances the classification ability of distilled images by incorporating an additional cross-entropy loss into the objective function. Furthermore, IDM introduces a mechanism of constructing a model pool to maintain diversity among the models used for distillation, alleviating the potential issue of overfitting to specific features or representations.

**Mixture-of-Experts.** Building upon multiple expert sub-networks in which each focuses on its specific sub-task or sub-domain within the overall problem space, the Mixture-of-Experts (MoE) architecture (Jacobs et al., 1991; Jordan & Jacobs, 1993; Zhou et al., 2020; Shazeer et al., 2017; Fedus et al., 2022) presents a compelling approach to leverage the specialized knowledge from these sub-networks within a single and unified model, where a gating module dynamically routes input data to the most relevant expert based on its characteristics, allowing the model to adapt to diverse inputs and learn distinct representations. In this paper, we adapt the core idea of MoE analogously to the problem of dataset distillation, where multiple experts of dataset distillation are encouraged to capture the characteristics of the original dataset from diverse perspectives, making the resultant distilled dataset more generalizable for further use of training models of various network architectures.

## 3 METHODOLOGY

### 3.1 PRELIMINARY

**Dataset Distillation.** Given a dataset $\mathcal{T} = \{(\boldsymbol{x}_i, \boldsymbol{y}_i)\}_{i=1}^{|\mathcal{T}|}$ composed of $|\mathcal{T}|$ real images $\boldsymbol{x}$ and their corresponding labels $\boldsymbol{y}$, the main goal of data distillation is to compress $\mathcal{T}$ into a learnable synthetic dataset $\mathcal{S} = \{(\boldsymbol{x}_i', \boldsymbol{y}_i')\}_{i=1}^{|\mathcal{S}|}$ with $|\mathcal{S}| \ll |\mathcal{T}|$, while attempting to encapsulate the knowledge of $\mathcal{T}$ in $\mathcal{S}$. With taking a model parameterized by $\theta$ as the bridge, the learning of such synthetic dataset is typically realized by minimizing the discrepancy between the model's behavior when being trained on $\mathcal{T}$ and $\mathcal{S}$ (i.e. the model trained on $\mathcal{S}$ ideally should have comparable performance with the one trained on $\mathcal{T}$) and can be generally written as:

$$\mathcal{S} = \arg\min_{\mathcal{S}} \mathcal{L}(\mathcal{S}, \mathcal{T}), \tag{1}$$

where $\mathcal{L}$ can follow various forms according to the particular designs in different distillation approaches. As described in previous sections, there are three major categories of dataset distillation methods where in this paper we adopt one representative approach per category to conduct our experiments and studies, i.e. IDC (Kim et al., 2022) for gradient matching, MTT (Cazenavette et al.,

Figure 1: The overall workflow of our proposed Mixture-o-Experts framework for Dataset Distillation. Initially, we pretrain and store multiple model parameters. Then each expert initializes its synthetic data from the original full dataset. During the distillation stage, each expert updates synthetic data by minimizing distance correlation between experts to promote diversity within whole synthetic data. Finally, after the distillation is finished, we apply mixup-based fusion among experts to enhance the transferability and performance of the distilled dataset when training a new model for further applications.

2022) for trajectory matching, and IDM (Zhao et al., 2023) for distribution matching, respectively. Here we briefly review their core designs for building $\mathcal{L}$.

**IDC (Kim et al., 2022)** aims to match the gradients produced by real data or distilled synthetic data with respect to the model parameters. Basically, given a model $\phi$ parameterized by $\theta$ which has been trained on the original dataset for a limited number of epochs, with denoting the real data samples and synthetic data samples of the same class $c$ as $Q_c^{\mathcal{T}}$ and $Q_c^{\mathcal{S}}$ respectively, IDC realizes the gradient matching via the following objective:

$$\mathcal{L}_{\text{IDC}} = \sum_{c=1}^{C} \sqrt{\left(\frac{1}{|Q_c^{\mathcal{T}}|} \sum_{(x_i, y_i) \in Q_c^{\mathcal{T}}} \nabla_\theta \ell(\phi_\theta(x_i), y_i) - \frac{1}{|Q_c^{\mathcal{S}}|} \sum_{(x_i', y_i') \in Q_c^{\mathcal{S}}} \nabla_\theta \ell(\phi_\theta(x_i'), y_i'))^2} \quad (2)$$

where $C$ denotes the number of classes and $\ell$ denotes the cross-entropy loss function.

**MTT (Cazenavette et al., 2022)** aims to match the long-range trajectories of updating model parameters between using real data or distilled synthetic data to train the model. Basically, with the same initial model parameters $\theta$, the model variant which is trained upon synthetic data for $N$ steps (resulting to have model parameter $\hat{\theta}_N$) is encouraged to match another variant which is trained upon real data for $M$ steps (resulting to have model parameter $\tilde{\theta}_M$) in terms of parameters, where $N \ll M$. The objective is formulated as:

$$\mathcal{L}_{\text{MTT}} = \frac{\|\hat{\theta}_N - \tilde{\theta}_M\|}{\|\theta - \tilde{\theta}_M\|} \quad (3)$$

where the denominator is independent to the distilled data and serves as a normalized term.

**IDM (Zhao et al., 2023)** aims to match the feature distribution of real dataset and the one of the distilled synthetic data (where the feature is extracted by a model $\phi$ parameterized by $\theta$ which has been trained on the original dataset) in which the metric for measuring the distance between two distributions is maximum mean discrepancy (MMD (Gretton et al., 2012)). Moreover, IDM has an additional regularization term for encouraging the distilled synthetic data to have minimal classification error. The overall objective for IDM is hence formulated as:

$$\mathcal{L}_{\text{IDM}} = \sum_{c=1}^{C} \|\frac{1}{|Q_c^{\mathcal{T}}|} \sum_{x_i \in Q_c^{\mathcal{T}}} \phi_\theta(x_i) - \frac{1}{|Q_c^{\mathcal{S}}|} \sum_{x_i' \in Q_c^{\mathcal{S}}} \phi_\theta(x_i')\| + \lambda \sum_{(x_i', y_i') \in \mathcal{S}} \ell(\phi_\theta(x_i'), y_i') \quad (4)$$

where $\lambda$ is to balance the distribution matching term and the regularization term.

**Distance Correlation.** Correlation analysis plays a crucial role in statistical analysis, allowing us to quantify the relationship between variables. While traditional correlation measures like Pear-

son's correlation coefficient focus on linear relationships, distance correlation offers a more versatile approach by capturing both linear and non-linear dependencies between variables. This makes it particularly well-suited for analyzing complex relationships often encountered in deep learning applications.

Let $(x_i, y_i)$, where $i = 1, 2, ..., n$, represent observed samples from the joint distribution of random variables $X$ and $Y$. To compute distance correlation, we first define the following quantities:

$$a_{j,k} = \|x_j - x_k\|, \quad \overline{a}_{j,\cdot} = \frac{1}{n}\sum_{m=1}^{n} a_{j,m}, \quad \overline{a}_{\cdot,k} = \frac{1}{n}\sum_{m=1}^{n} a_{m,k}, \quad \overline{a}_{\cdot,\cdot} = \frac{1}{n^2}\sum_{j,k=1}^{n} a_{j,k}$$

$$b_{j,k} = \|y_j - y_k\|, \quad \overline{b}_{j,\cdot} = \frac{1}{n}\sum_{m=1}^{n} b_{j,m}, \quad \overline{b}_{\cdot,k} = \frac{1}{n}\sum_{m=1}^{n} b_{m,k}, \quad \overline{b}_{\cdot,\cdot} = \frac{1}{n^2}\sum_{j,k=1}^{n} b_{j,k}$$

$$A_{j,k} = a_{j,k} - \overline{a}_{j,\cdot} - \overline{a}_{\cdot,k} + \overline{a}_{\cdot,\cdot}, \qquad B_{j,k} = b_{j,k} - \overline{b}_{j,\cdot} - \overline{b}_{\cdot,k} + \overline{b}_{\cdot,\cdot}$$

where $j, k = 1, 2, ..., n$ and $n$ is the total number of samples. These quantities represent pairwise distances between samples and their respective averages.

The squared sample distance covariance is then calculated as the average product of $A_{j,k}$ and $B_{j,k}$:

$$dCov_n^2(X, Y) = \frac{1}{n^2}\sum_{j=1}^{n}\sum_{k=1}^{n} A_{j,k}B_{j,k}, \quad dVar_n^2(X) = dCov_n^2(X, X) = \frac{1}{n^2}\sum_{j,k} A_{j,k}^2$$

Finally, the distance correlation between $X$ and $Y$ is defined as:

$$dCor^2(X, Y) = dCov^2(X, Y)/\sqrt{dVar^2(X)dVar^2(Y)},$$

This value ranges from 0 to 1, where 0 indicates independence and 1 indicates perfect dependence between the variables. In the context of dataset distillation, distance correlation serves as a valuable tool for assessing the diversity of the generated synthetic data. By minimizing the distance correlation between expert models trained on distilled data, we encourage the models to learn distinct representations, thus promoting a more diverse and informative synthetic dataset.

## 3.2 DATASET DISTILLATION WITH MULTIPLE EXPERTS

As motivated previously, the existing dataset distillation methods often suffers from poor cross-architecture generalization (i.e. when the distilled dataset is adopted to train a model that has different network architecture from the one used in the distillation process, the performance of the resultant model degrades significantly) and we attribute such problem to their relying on a "single" distillation process for encapsulating the entire original dataset, where the distilled dataset might not have sufficient diversity to support various architectures. To this end, we propose to leverage the mixture-of-experts idea for having multiple distillation processes (as experts) in a unified framework and particularly encourage them to capture distinct characteristics of the original data thus enriching the diversity of the resultant distilled dataset as well as improving the cross-architecture generalizability.

In detail, we introduce $K$ experts $\mathbb{E} = \{\mathcal{E}_1, \mathcal{E}_2, ..., \mathcal{E}_K\}$, where each expert $\mathcal{E}_i$ is tasked to produce a specific subset $\mathcal{S}_i$ of synthetic dataset $\mathcal{S}$, derived from its corresponding subset $\mathcal{T}_i$ of real data $\mathcal{T}$. Noting that $\mathcal{S} = \mathcal{S}_1 \cup \mathcal{S}_2 \cup \cdots \cup \mathcal{S}_K$ and $\mathcal{T} = \mathcal{T}_1 \cup \mathcal{T}_2 \cup \cdots \cup \mathcal{T}_K$ with setting $\mathcal{T}_i \bigcap \mathcal{T}_j = \varnothing, \ \forall i \neq j$ (for simplicity, we assume $|\mathcal{T}|$ is divisible by $K$). Particularly, in addition to having each expert independently perform dataset distillation on its assigned subset, we advance to introduce a loss function based on distance correlation (Zhen et al., 2022; Székely et al., 2007) which minimizes the dependency between the feature representations of synthetic subsets distilled by different experts, thus preventing the experts from converging towards the similar synthetic subsets (in terms of feature representations) as well as prompting the diversity in the distilled dataset. Noting that the distance correlation was proposed by (Székely et al., 2007) almost two decades ago but is recently revisited by (Zhen et al., 2022) to highlight its advantages and general uses in deep learning as a measure for comparing two feature representations/spaces or the functional behavior between two networks. In implementation, once every few iterations during the distillation process, we randomly select pairs of synthetic subsets (e.g. $\mathcal{S}_i$ and $\mathcal{S}_j$ distilled by two distinct experts $\mathcal{E}_i$ and $\mathcal{E}_j$ respectively) and for

each pair we compute its distance correlation. For a pair of synthetic subsets ($\mathcal{S}_i$ and $\mathcal{S}_j$), our loss based on distance correlation is written as:

$$\mathcal{L}_{Corr} = \texttt{dCor}^2(\phi(\mathcal{S}_i), \phi(\mathcal{S}_j)) \tag{5}$$

where $\phi$ is a model which has been pretrained on real data and used for feature extraction, and $\texttt{dCor}^2(\cdot, \cdot)$ is the squared sample distance covariance (Zhen et al., 2022; Székely et al., 2007).

The overall objective combines the individual distillation losses of each expert (e.g. $\mathcal{L}_{\text{IDC}}$, $\mathcal{L}_{\text{MTT}}$, or $\mathcal{L}_{\text{IDM}}$, depending on which distillation methods are used to construct our experts) and the loss based on distance correlation:

$$\mathcal{L} = \sum_{i=1}^{K} \mathcal{L}_{DD}(\mathcal{S}_i; \mathcal{T}_i) + \sum_{i=1}^{K} \sum_{j \neq i}^{K} \mathcal{L}_{Corr}(\mathcal{S}_i, \mathcal{S}_j; \phi), \tag{6}$$

where $\mathcal{L}_{DD}(\mathcal{S}_i; \mathcal{T}_i)$ represents the distillation loss (which could be $\mathcal{L}_{\text{MTT}}$, $\mathcal{L}_{\text{IDC}}$ or $\mathcal{L}_{\text{IDM}}$) according to the distillation method used for the expert $\mathcal{E}_i$, operating on its assigned real data subset $\mathcal{T}_i$ and generating the synthetic subset $\mathcal{S}_i$.

### 3.3 MIXUP-BASED FUSION

While the distilled synthetic images can be directly utilized for downstream tasks, this approach may compromise overall accuracy due to the potential for specialized representations learned by individual expert models. Although minimizing distance correlation effectively promotes diversity within the distilled dataset, it may inadvertently lead to experts focusing on limited aspects of the original data distribution. To address this challenge and encourage a more holistic representation, we incorporate a fusion strategy inspired by Mixup (Zhang et al., 2018). During the evaluation phase, we combine synthetic images from different expert subsets using a weighted sum:

$$\hat{x}' = \lambda x'_p + (1 - \lambda)x'_q \tag{7}$$

where $x'_p \in S_i$, $x'_q \in S_j$, $p = 1, \ldots, |S_i|, q = 1, \ldots, |S_j|$ and $\forall i \neq j$. Here, $\lambda$ is a scalar value sampled from a Beta distribution, following the principles outlined in the original Mixup (Zhang et al., 2018) approach. This Mixup-based fusion encourages the model to learn from the combined knowledge of different experts, effectively leveraging their specialized representations to create a more generalizable and comprehensive representation of the original data. This approach aims to improve both the accuracy of downstream tasks and the model's ability to adapt to diverse network architectures.

## 4 EXPERIMENTAL RESULTS

This section presents the evaluation of our proposed framework in conjunction with three dataset distillation methods. We compare the performance of our approach to the original methods, highlighting the benefits of incorporating multiple expert assistants and promoting diversity within the distilled dataset.

### 4.1 EXPERIMENTAL SETUP

**Datasets.** To comprehensively evaluate our multi-expert DD framework, we employed a diverse set of datasets with varying image resolutions and complexities. Our benchmark for low-resolution datasets was CIFAR-10/100 (Krizhevsky, 2009), each containing 60,000 color images ($32 \times 32$) across 10 and 100 classes respectively. For higher resolutions, we utilized ImageNette, a 10-class subset of ImageNet (Deng et al., 2009), ImageWoof, focusing on 10 dog breeds, and STL-10 (Coates et al., 2011), a 10-class dataset with labeled and unlabeled images. By evaluating our framework across these datasets (all resized to $64 \times 64$), we demonstrate its efficacy and robustness for addressing the challenge of cross-architecture generalizability in dataset distillation task.

**Network Architectures.** To ensure consistency with prior works on dataset distillation (Zhao & Bilen, 2021b; Kim et al., 2022; Zhao et al., 2023; Cazenavette et al., 2022), we initially employed a ConvNet-3 architecture for the distillation task. This architecture, a commonly used baseline in

dataset distillation research, consists of three identical convolutional blocks followed by a linear classifier. Each block comprises a $3 \times 3$ convolutional layer with 128 kernels, instance normalization, ReLU activation, and a $3 \times 3$ average pooling layer with a stride of 2. This configuration enables efficient feature extraction while maintaining a manageable model complexity. To accommodate higher-resolution datasets, such as ImageNette and STL-10, we augmented the ConvNet-3 architecture by adding a fourth convolutional block. This addition allows for the capture of richer image details present in higher-resolution images, enabling improved feature representation and downstream performance. For the cross-architecture evaluation of our framework, we employed three established architectures: ResNet-18 (He et al., 2016), VGG-11 (Simonyan & Zisserman, 2015), and AlexNet (Krizhevsky et al., 2012). These architectures represent a diverse set of network designs, each with distinct architectural features and computational complexities. By evaluating the performance of our framework across these architectures, we aim to demonstrate its generalizability and robustness in transferring distilled knowledge to a range of network configurations.

**Baselines.** To showcase the versatility of our proposed framework, we integrate it with three prominent surrogate objectives for dataset distillation: IDC (Kim et al., 2022), IDM (Zhao et al., 2023), and MTT (Cazenavette et al., 2022), representing gradient matching, distribution matching, and trajectory matching, respectively. We conduct a comparative analysis of these distillation methods under equivalent storage budget, which means the same images-per-class times the number-of-experts (IPC $\times$ NoE), constraining and reporting their respective cross-architecture performance.

**Implementation Details.** As mentioned previously, three dataset distillation methods (i.e. IDC (Kim et al., 2022), IDM (Zhao et al., 2023), and MTT (Cazenavette et al., 2022)) are leveraged for integration with our proposed frameworks: For IDC, we adopt its optimization strategy which decouples the synthetic dataset and the model parameters while omitting the multi-formation aspect to maintain consistency and focus on the core distillation process; For IDM, we employ its loss function which incorporates Maximum Mean Discrepancy (MMD) loss to align the distribution of synthetic data with the real data, along with a cross-entropy loss term to enhance the classification ability of the synthetic data. To avoid complexity and ensure compatibility with our multi-expert framework, we excluded the model queue technique; For MTT, we maintain a fixed learning rate for the synthetic data across all expert models, ensuring consistency in the learning process and facilitating the subsequent fusion and classification stages. Prior to distillation, we trained and saved over 100 model trajectories for each dataset. We generated synthetic images using 10 and 20 Images Per Class (IPC) from the real datasets. Within our framework, we split the total IPC equally among experts and compared their performance to single-expert baselines. To optimize the synthetic images generated for CIFAR-10 and CIFAR-100, we employed the SGD optimizer with a fixed learning rate and a momentum, following prior works (Zhao et al., 2023; Kim et al., 2022; Cazenavette et al., 2022). We incorporated differentiable augmentation strategies during both the learning and evaluation phases, a common practice in dataset distillation. To encourage diverse learning, we utilized different subsets of real data for IDC and IDM, selecting samples based on low classification losses (i.e., easy samples). For our fusion strategy, we employed a weighted sum of synthetic images from different expert models, where the weights were sampled from a Beta distribution with parameters set to 0.5. To facilitate a rapid assessment of the learning progress, we limited the distillation process for low- and medium-resolution images to 100 iterations. For how the subsets are selected and utilized, we select real data subsets based on the classification loss predicted by the pre-trained model, i.e. we specifically record the loss value for each real data sample. These loss values are then used to rank the difficulty of classifying each sample. We subsequently select the top 10% to 30% of samples with the lowest loss values, which represent "easy samples" that are readily classified by the model. These easy samples are then used to form the synthetic subsets $\mathcal{S}_i$ (as initialization) assigned to each expert during the distillation process.

## 4.2 COMPARISON BETWEEN SINGLE EXPERT AND MULTIPLE EXPERTS

We conducted an experiment to evaluate the impact of multiple experts using three different distillation methods: IDC, IDM, and MTT, under two distinct storage budgets: 10 and 20. To compare the effects of multiple experts, we established two scenarios for each storage budget: allocating the entire budget to a single expert and equally dividing the budget between two experts. Specifically, for a storage budget of 10, we set the number of experts (NoE) to 1 with images per class (IPC) being 10, denoted by IPC $\times$ NoE = $10 \times 1$, and to 2 with IPC being 5 for each expert, denoted by IPC $\times$ NoE = $5 \times 2$. Similarly, for a storage budget of 20, we set IPC $\times$ NoE to $20 \times 1$ and $10 \times 2$.

Table 1: Distillation and cross-architecture performance on CIFAR-10/100 datasets. Results compare single-expert baselines (NoE = 1) with our proposed multi-expert framework (NoE > 1) using IDM (Zhao et al., 2023), IDC (Kim et al., 2022) and MTT (Cazenavette et al., 2022) distillation methods under the same storage budget. We use ConvNet-3 to perform distillation. IPC and NoE stand for the "images per class" and "number of experts" respectively.

| Target | CIFAR-10 | | | | CIFAR-100 | | | |
|---|---|---|---|---|---|---|---|---|
| Budget | 10 | | 20 | | 10 | | 20 | |
| IPC×NoE | 5×2 | 10×1 | 10×2 | 20×1 | 5×2 | 10×1 | 10×2 | 20×1 |
| IDC | | | | | | | | |
| ConvNet-3 | **53.32** | 52.73 | **55.52** | 54.05 | **31.44** | 31.03 | **36.03** | 35.33 |
| VGG11 | **51.41** | 49.70 | **53.37** | 51.74 | **29.62** | 27.38 | **34.53** | 31.25 |
| ResNet18 | **51.56** | 47.51 | **54.06** | 50.06 | **30.74** | 28.59 | **36.08** | 34.09 |
| AlexNet | **46.34** | 40.69 | **52.01** | 44.04 | **26.45** | 20.05 | **33.84** | 25.76 |
| IDM | | | | | | | | |
| ConvNet-3 | **49.04** | 47.61 | **52.31** | 50.48 | 26.87 | **27.45** | **32.38** | 31.67 |
| VGG11 | 44.53 | **44.65** | **48.31** | 47.70 | **23.06** | 22.90 | 28.57 | **28.84** |
| ResNet18 | **45.05** | 42.84 | **49.52** | 46.02 | 24.29 | 25.35 | **32.08** | 32.04 |
| AlexNet | **37.19** | 34.96 | **44.32** | 41.71 | **19.49** | 17.07 | **27.16** | 22.59 |
| MTT | | | | | | | | |
| ConvNet-3 | 53.82 | **56.17** | 60.14 | **60.82** | **30.00** | 29.53 | 28.50 | **29.80** |
| VGG11 | **49.86** | 46.59 | **52.04** | 47.27 | **29.23** | 27.99 | **27.80** | 27.72 |
| ResNet18 | **52.14** | 46.17 | **58.13** | 53.94 | **30.19** | 27.81 | **27.55** | 26.05 |
| AlexNet | **44.16** | 32.97 | **52.11** | 50.01 | **25.27** | 23.88 | **28.10** | 27.67 |
| Full Dataset | 84.80 | | | | 56.20 | | | |

The distillation process was performed on ConvNet-3. After completing the distillation, we trained new networks, including ConvNet-3, VGG11, ResNet18, and AlexNet, on the distilled dataset to measure the performance of the distillation methods. Table 1 presents the experimental results on CIFAR-10 and CIFAR-100. We observed consistent performance improvements in multi-experts (i.e. NoE equals to 2) built by IDC over single-expert (i.e. NoE equals to 1) baselines, and most of the multi-expert results built by IDM and MTT outperformed the single-expert baselines. This suggests that the distilled datasets generated by our multi-expert framework are more generalizable for new architectures.

## 4.3 ABLATION STUDIES

### 4.3.1 IMPORTANCE OF DISTANCE CORRELATION AND MIXUP-BASED FUSION.

Table 2 presents the findings of an ablation study designed to evaluate the individual contributions of two key components within our proposed framework: distance correlation minimization and mixup-based fusion. We analyze their impact on performance using the CIFAR-10 dataset and employ IDM (Zhao et al., 2023), IDC (Kim et al., 2022) and MTT (Cazenavette et al., 2022) as the surrogate distillation methods and varying IPC settings. Incorporating distance correlation generally leads to a improvement in performance across all three distillation methods and IPC settings. This suggests that encouraging diversity among expert models through distance correlation helps capture a broader range of information from the original data, leading to better generalizability. However, the magnitude of the improvement varies depending on the specific distillation method and IPC level. The inclusion of the mixup-based fusion strategy improves performance across all distillation methods and IPC settings. This highlights the effectiveness of combining complementary information learned by different expert models. Finally, the combination of distance correlation and mixup-based fusion generally results in the best overall performance using the MoE framework.

### 4.3.2 EVALUATING THE IMPACT OF MIXUP-BASED FUSION.

Table 3 presents an ablation study investigating the efficacy of our proposed mixup-based fusion in comparison to the vanilla mixup technique (Zhang et al., 2018) within the context of dataset distillation using IDC (Kim et al., 2022) as the surrogate method. The results demonstrate that our strategy, which mixes synthetic images from different expert models (c.f. Section 3.3), consistently outper-

Table 2: Ablation study on our proposed distance correlation and fusion techniques on CIFAR-10 with using different dataset distillation methods to build experts (e.g. IDM (Zhao et al., 2023), IDC (Kim et al., 2022) and MTT (Cazenavette et al., 2022)).

| Dataset | CIFAR-10 | | | | | |
|---|---|---|---|---|---|---|
| Distillation Method | IDM | | IDC | | MTT | |
| IPC×NoE | 5×2 | 10×2 | 5×2 | 10×2 | 5×2 | 10×2 |
| Baseline | 46.35 | 49.50 | 52.28 | 53.47 | 51.02 | 58.08 |
| + Distance Correlation | 47.55 | 51.81 | 52.90 | 55.01 | 51.59 | 58.38 |
| + Distance Correlation & Mixup-based Fusion | 49.04 | 52.31 | 53.32 | 55.52 | 53.82 | 60.14 |

Table 3: Ablation study on using different mixup strategies. Mixup is applied while training new network architectures on the distilled dataset. The distilled dataset is built using IDC on CIFAR-10 with IPC×NoE = $5 \times 2$. "Ours" refers to our proposed method, which mixes images from different experts, as introduced in Section 3.3. "Vanilla Mixup" means mixing the distilled images without considering the constraint of different experts. "w/o Mixup" involves training a new network on the distilled dataset without using any mixup.

| Fusion strategy | Ours | Vanilla Mixup | w/o Mixup |
|---|---|---|---|
| ConvNet-3 | **53.32** | 52.46 | 53.21 |
| VGG11 | **51.41** | 51.34 | 50.84 |
| ResNet18 | **51.56** | 50.20 | 50.11 |
| AlexNet | **46.34** | 44.60 | 42.98 |

forms both the baseline without mixup and the vanilla mixup approach (mixing images from both the same expert and different experts) across all target architectures. This suggests that leveraging the diverse representations learned by multiple experts is crucial for enhancing the generalizability and performance of the distilled dataset. While vanilla mixup also exhibits some performance improvement over the baseline, it falls short of our proposed strategy, indicating that simply mixing images may not sufficiently capture the full range of information required for effective distillation.

## 4.4 IMPACT OF NUMBER OF EXPERTS

To investigate the influence of the number of experts on performance, we conducted experiments on CIFAR-10 using the IDM distillation method with a fixed total IPC of 30. We varied the number of experts, distributing the total IPC equally among them. Table 4 presents the results for one expert (30×1), two experts (15×2), and three experts (10×3) across different target architectures. As shown in the table, using two experts generally improves performance compared to a single expert. Furthermore, for some architectures like ConvNet-3 and ResNet18, increasing the number of experts to three leads to additional performance gains. However, the improvement from two to three experts is smaller than that from one to two experts, and in the case of VGG11 and AlexNet, the gains are marginal, or performance may even decrease. This suggests that while increasing the number of experts can enhance performance by capturing more diverse aspects of the data, there might be a point of diminishing returns. The optimal number of experts likely depends on factors such as the dataset, the distillation method, and the target architectures. Further investigation is needed to fully understand this trade-off and determine the optimal configuration for various scenarios.

## 4.5 SUPERVISED CONTRASTIVE LEARNING

DD aims to compress a large dataset while preserving its essential information for downstream tasks. A critical question arises: can distilled datasets effectively support other tasks? To investigate this, we evaluated the performance of Supervised Contrastive Learning (SupCon)(Khosla et al., 2020), both with and without our proposed multi-expert framework. Our analysis focused on a total IPC of 20 and assessed both the accuracy on the distillation task and the transfer performance to other datasets. To assess the effectiveness of our distilled dataset for unsupervised representation learning,

Table 4: Performance Comparison with Varying Number of Experts using IDM Distillation on CIFAR-10 (Total IPC=30)

| IPC×NoE | 30×1 | 15×2 | 10×3 |
|---|---|---|---|
| ConvNet-3 | 53.33% | 55.62% | **56.43%** |
| VGG11 | 52.53% | **53.26%** | 53.16% |
| ResNet18 | 51.91% | 54.75% | **55.10%** |
| AlexNet | 44.91% | 48.66% | **49.22%** |

Table 5: Supervised Contrastive Learning (SupCon) Performance on ImageNette Distilled using Single and Multi-Expert IDM. Results include SupCon accuracy on distilled ImageNette (Total IPC=20) and transfer learning performance on ImageWoof and STL-10.

| IPC×NoE | Test Acc. in Distillation | SupCon Acc. | Transfer Performance | |
|---|---|---|---|---|
| | | | ImageWoof | STL-10 |
| 20×1 | 49.98 | 34.57 | 14.63 | 21.08 |
| 10×2 | **53.27** | **46.62** | **17.12** | **29.02** |
| Full Dataset | - | 71.51 | 18.85 | 32.95 |

we trained a ConvNet-4 model on the distilled ImageNette dataset using SupCon. After training, we froze the feature extractor and fine-tuned only the classifier. The classification accuracy, denoted as "SupCon Acc." in Table 5, was evaluated by testing the model on the original ImageNette test dataset. This evaluation aimed to assess how effectively the distilled data facilitates the learning of robust feature representations, as evidenced by its performance on unseen test data. To further evaluate the generalizability of the learned feature representations, we performed a transfer learning experiment. We froze the feature extractor (backbone) of the SupCon model trained on the distilled ImageNette dataset and fine-tuned the final classification layer on the target datasets (ImageWoof and STL-10). The transfer performance is reported as the classification accuracy on these target datasets. Our proposed framework with two experts ($10 \times 2$) achieves a significantly higher SupCon accuracy ($46.62\%$) compared to the baseline IDM method with a single expert which achieves $34.57\%$. This indicates that the diversity and generalizability promoted by our framework lead to a distilled dataset that is more effective for unsupervised representation learning using SupCon. Moreover, the distilled dataset generated by our method also exhibits improved transfer performance across all target datasets compared to the baseline. While both distillation methods show a performance gap compared to training SupCon on the entire ImageNette dataset, our method significantly narrows this gap. This suggests that our framework effectively captures a substantial portion of the information present in the original data, even with a significantly reduced dataset size.

## 5 CONCLUSION

This work has presented a novel approach to dataset distillation that addresses the challenge of cross-architecture performance degradation by promoting diversity and generalizability within the distilled dataset. Our proposed framework leverages multiple expert models, each specializing in distilling a distinct subset of the data, and incorporates distance correlation minimization as well as image fusion strategies to enhance the richness and informativeness of the distilled data. Through extensive experiments, we have demonstrated the effectiveness of our approach in mitigating the transferability issue and achieving improved performance across various target architectures, particularly in low-data regimes. Our findings highlight the importance of diversity in dataset distillation and provide valuable insights for future research in this area.

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

# A APPENDIX

## A.1 PERFORMANCE WITH HIGHER IPC

To investigate the impact of a higher IPC (Images Per Class) on performance, we conducted experiments on CIFAR-10 using the IDM distillation method with a total IPC of 50. We compared the performance of our multi-expert framework with two experts (25 images per class per expert, denoted as 25×2) against a single-expert baseline (50×1). The results are presented in Table 6.

The table shows that our multi-expert framework achieves comparable or superior performance to the single-expert baseline across different target architectures. Notably, for ConvNet-3, ResNet18, and AlexNet, the multi-expert framework demonstrates a clear advantage. This indicates that even with a higher IPC, our approach can effectively leverage the diverse representations learned by multiple experts to improve performance. However, for VGG11, the single-expert baseline performs slightly better. This might indicate that the optimal configuration of our framework might vary depending on the specific target architecture and potentially, the relative gain of the multi-expert framework may start to diminish when IPC is increased to have less overfitting effect.

## A.2 PERFORMANCE ON LARGER DATASET

To evaluate the scalability and effectiveness of our multi-expert framework on a more complex dataset, we conducted experiments on TinyImageNet. This dataset comprises 200 classes and images with a higher resolution ($64 \times 64$) compared to CIFAR-10, presenting a more challenging

| IPC×NoE | 25×2 | 50×1 |
|---|---|---|
| ConvNet-3 | **59.57** | 55.43 |
| VGG11 | 56.00 | **58.00** |
| ResNet18 | **59.10** | 57.22 |
| AlexNet | **54.36** | 50.79 |

| IPC×NoE | 20×1 | 10×2 |
|---|---|---|
| ConvNet-4 | 13.51% | **18.39%** |
| VGG11 | 13.86% | **15.04%** |
| ResNet18 | 11.48% | **14.97%** |
| AlexNet | 9.18% | **12.29%** |

Table 6: Comparison of Single-Expert and Multi-Expert IDM Distillation on CIFAR-10 (Total IPC=50)

Table 7: Comparison of IDM Performance on TinyImageNet with Single and Multi-Expert Distillation (Total IPC=20)

distillation task. We used the IDM distillation method with a total IPC of 20 and compared the performance of our multi-expert framework (10×2) against a single-expert baseline (20×1) across various target architectures, including ConvNet-4, VGG11, ResNet18, and AlexNet. The results are presented in Table 7.

As shown in the table, our multi-expert framework consistently outperforms the single-expert baseline across all evaluated architectures. These results demonstrate the effectiveness and scalability of our approach on a more challenging dataset with a larger number of classes and higher image resolution. The significant performance improvement observed on TinyImageNet further strengthens our claim that the multi-expert framework can enhance dataset distillation across diverse dataset characteristics.

### A.3    EXPLORING DIFFERENT ARCHITECTURES FOR EXPERTS

To investigate the potential of using different architectures for the experts in our multi-expert framework, we conducted an experiment using a combination of ConvNet-3 and ResNet18. One potential avenue for improving cross-architecture generalization in dataset distillation, besides enhancing data diversity, is to incorporate diverse model architectures during the distillation process. This approach aims to capture a broader range of feature representations and learn more generalizable distilled datasets. In our experiment, we employed the IDM distillation method with a total IPC of 10, distributed evenly among the two experts (5×2). The results are presented in Table 8.

While using two different architectures (ConvNet-3 and ResNet18) for experts did not lead to significantly better performance compared to using two ConvNet-3 models across all target architectures, the performance degradation with different architectures is slightly smaller. For instance, while both configurations show a performance drop for AlexNet when trained on the distilled dataset, the drop is more pronounced for the case where both experts are ConvNet-3. These results suggest that using different architectures for experts may potentially reduce overfitting to a specific architecture, although further research is needed to fully understand this effect. This approach holds promise for future investigations aimed at directly improving cross-architecture generalization.

Table 8: Impact of Expert Architectures on Distillation Performance (IDM, Total IPC=10)

| Models | 2 × ConvNet-3 | 1 × ConvNet-3 1 × ResNet18 |
|---|---|---|
| IPC×NoE | 5×2 | |
| ConvNet-3 | 49.04% | 42.94% |
| VGG11 | 44.53% (↓ 9.1%) | 39.53% (↓ 7.9%) |
| ResNet18 | 45.05% (↓ 8.1%) | 39.80% (↓ 7.3%) |
| AlexNet | 37.19% (↓ 24.1%) | 34.83% (↓ 18.8%) |

### A.4    VISUALIZATION

To visually evaluate the quality of synthetic images generated by our multi-expert framework, we compared it to the single-expert IDM (Zhao et al., 2023) method using CIFAR-10 (Krizhevsky, 2009) and IDM as the distillation method. Figure 2 shows a comparison of synthetic images distilled

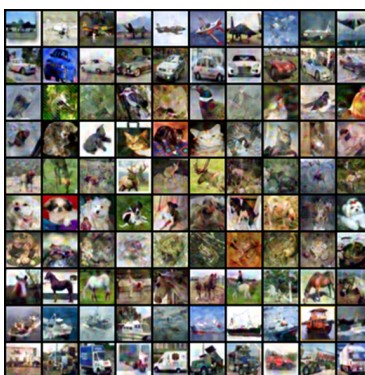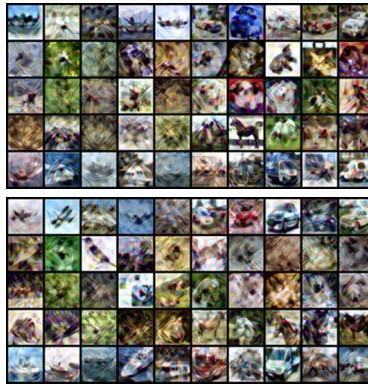

Figure 2: Left: Distilled images from a single expert trained on CIFAR-10 using IDM (IPC=10). Right: Two sets of distilled images, each generated by a separate expert using IDM on CIFAR-10 (IPC×NoE = 5×2), illustrating the diversity introduced by our multi-expert framework.

under identical settings (iteration counts and total IPC): IPC=5 with two experts for the multi-expert framework (total IPC=10) and IPC=10 for the single-expert baseline. Our observations reveal that synthetic images generated by the single-expert approach (left) exhibit less detail in object textures and shapes. Conversely, synthetic images produced by our multi-expert framework (right) demonstrate greater fidelity to the original data, with more accurate representations of textures and shapes. This visual evidence supports our claim that our framework promotes diversity and richer information captured within the distilled dataset, leading to the generation of more realistic and informative synthetic images.

