# OpenReview forum: "Four eyes see more than two: Dataset Distillation with Mixture-of-Experts"
_ICLR.cc/2025/Conference — ICLR 2025 Conference Withdrawn Submission_

### Official Review · Reviewer_4Jux · 2024-10-28

**Soundness:** 2
**Presentation:** 3
**Contribution:** 2
**Rating:** 5
**Confidence:** 4

**Summary:**

This paper introduces a MOE framework for dataset distillation, which applies multiple experts to distill synthetic images in multiple views. During distillation, a distance correlation metric is utilized to improve the diversity of the images from multiple experts. During the evaluation phrase, authors merge the images from multiple experts by applying mix up.

**Strengths:**

1. The proposed method seems to be compatible with most of the previous dataset distillation methods since the images from each expert can be obtained in any distillation method.
2. Authors claim that they achieve better cross-architecture performance for dataset distillation.
3. Authors give a detailed introduction to their implementation details. (It will be better if codes are released).

**Weaknesses:**

1. The first problem is whether the MOE framework really brings benefits. In Table 2, the authors give the ablation study for using distance correlation and the mixup-based fusion. The setting that Mixup-based fusion is utilized while distance correlation is not utilized is not reported, while is important to prove the effectiveness of the multiple experts.
2, The ablation study can be further improved by the following experiments: using one expert for distillation with IPC=20, and performing fusion from  2 to 1, resulting in an IPC=10, comparing this result with the IPC=10 fused from two experts.
3. Authors mainly show performance improvements in cross-architecture experiments. How does this method work for models in the same architecture? Besides, why does such a MOE framework improve cross-architecture results but not work very well for the same architecture? It seems that the methodology of this work has no direct relation to cross-architecture experiments.
4. Some typos. Line 508.5, quotes are wrong.
5. CIFAR100 experiments are missing (not a very big problem).
6. It's really difficult for people to capture the idea of "diversity" from images in Figure 2. Why not provide images in high resolution such as ImageNet?

**Questions:**

Please refer to the weakness. 1-3 are my main concerns.

---

### Official Review · Reviewer_Amis · 2024-10-31

**Soundness:** 2
**Presentation:** 2
**Contribution:** 2
**Rating:** 5
**Confidence:** 5

**Summary:**

This paper proposes a dataset distillation method using a Mixture-of-Experts (MoE) framework to improve cross-architecture performance. By dividing the dataset distillation process across multiple expert models, each focusing on different data subsets, the approach aims to enhance data diversity and reduce architecture-specific overfitting.

**Strengths:**

- The paper addresses a relevant challenge in dataset distillation, proposing a ”mixture-of-experts" approach to improve cross-architecture performance and reduce model-specific overfitting.

- The method combines known techniques such as distance correlation minimization and Mixup fusion to enhance data diversity, demonstrating promising results in low-data regimes.

- Experimental results on multiple datasets indicate some potential benefits of using multiple experts to improve generalization across different architectures.

**Weaknesses:**

- Limited Practicality of Dataset Distillation Methods: While dataset distillation aims to reduce training costs, current methods—including this one—often require **significantly more computation to create distilled datasets than training directly on the full dataset.** This undermines the primary goal of efficiency, which the paper does not address by comparing training time and resource use with traditional methods.

- Lack of Novelty: The approach lacks substantive innovation, **primarily combining existing techniques** (MoE, distance correlation, Mixup) without substantial theoretical or methodological advances. The MoE structure used here is a **simple model ensemble** rather than a true sparsely activated mixture of experts with adaptive routing, limiting its contribution to the field.

- Outdated Baselines and Datasets: The experiments primarily use small datasets (e.g., CIFAR-10/100) and outdated models (e.g., ConvNet-3, VGG11), which do not adequately demonstrate the method's effectiveness or scalability on modern, large-scale tasks. In 2024, using such datasets does not reflect practical applicability, especially when larger datasets can be handled efficiently on current hardware.

**Questions:**

- How much computation is required to train a ConvNet-3 on the full CIFAR-100 dataset compared to the distillation and training process? Could the authors provide specific numbers to illustrate this comparison?

- Would using different architectures to generate distilled subsets yield better results than using the same architecture across subsets?

- What are the computational and time costs for distillation on larger datasets like ImageNet, and how do these compare to training directly on the full dataset?

- How does the proposed multi-expert approach scale with larger datasets and more complex architectures? Is there a significant increase in overhead?

- Could the authors provide a breakdown of resource usage and time for each stage of the distillation process? This would help evaluate its efficiency relative to traditional training.

---

### Official Review · Reviewer_rpr9 · 2024-11-03

**Soundness:** 2
**Presentation:** 3
**Contribution:** 2
**Rating:** 5
**Confidence:** 4

**Summary:**

This paper introduces a novel framework for dataset distillation using a mixture-of-experts (MoE) approach aimed at improving cross-architecture generalizability. The method, called Four Eyes See More Than Two, assigns different parts of the dataset distillation task to multiple expert models, each trained to distill a distinct subset of the data. To further promote diversity, the authors employ a distance correlation minimization strategy, encouraging each expert to capture unique data representations. Finally, the mixup-based fusion technique integrates the synthetic data from different experts, creating a more comprehensive distilled dataset for training across diverse architectures.

**Strengths:**

### Strengths
1. The paper tackles a challenging and relevant problem in dataset distillation, specifically addressing cross-architecture performance degradation. The application of mixture-of-experts, along with distance correlation minimization, is an innovative approach to promoting diversity in distilled data.

2. The authors conduct extensive experiments across various datasets (e.g., CIFAR-10, CIFAR-100) and network architectures (e.g., ConvNet-3, ResNet18, VGG-11, AlexNet). The results consistently demonstrate the advantages of the multi-expert framework in improving the performance and generalizability of the distilled datasets, especially under low-data conditions.

3. The mixup fusion strategy is thoughtfully designed and implemented to leverage complementary information across experts, enhancing the generalizability of the synthetic dataset.

4. The paper is generally well-organized, with mathematical formulations and a clear explanation of the methodology.

**Weaknesses:**

1. While distance correlation minimization is used as a diversity-promoting technique, there is limited theoretical justification as to why this particular metric would enhance generalizability across architectures. A stronger theoretical or empirical explanation connecting distance correlation with cross-architecture transferability would be beneficial.

2. Although the MoE approach is shown to improve generalization, it requires training and maintaining multiple experts. The paper lacks a discussion on the computational trade-offs involved and how they may impact scalability, particularly when larger datasets or more complex architectures are involved.

3. While the proposed method performs well, it would be insightful to compare the performance against more traditional regularization techniques or other diversity-promoting methods to strengthen the uniqueness of the approach.

4. The paper assumes an equal storage budget per expert, which might not always be practical. An analysis of how unequal storage distribution among experts impacts performance could add depth to the evaluation.

**Questions:**

1. It is recommended to provide more theoretical insights or empirical analysis on how distance correlation minimization specifically enhances generalizability across different architectures.

2. How does the computational cost of training multiple experts compare to a single-expert setup, and are there any optimizations that could make this approach more efficient?

3. Are there any alternative diversity-promoting strategies, such as ensemble regularization techniques, and if so, how do they compare to distance correlation minimization?

4. In scenarios where the storage budget is not equal across experts, how does this affect performance, please explore the impact of imbalanced storage distributions.

5. Please provide empirical evidence showing how distance correlation relates to feature diversity across architectures, or conduct ablation studies comparing distance correlation to other diversity metrics.

6. It is suggested to provide concrete metrics on training time and memory usage for their method compared to single-expert baselines, especially as the number of experts or dataset size increases.

7.  It is better to compare performance with different ratios of storage allocation between experts (e.g., 70-30 split vs 50-50) on a particular dataset.

8. It is suggested to include experiments with transformer architectures (e.g., ViT) or larger CNNs (e.g., EfficientNet) to demonstrate the method's generalizability across a wider range of modern architectures.

---

### Official Review · Reviewer_XVYm · 2024-11-03

**Soundness:** 3
**Presentation:** 3
**Contribution:** 2
**Rating:** 5
**Confidence:** 4

**Summary:**

This paper introduces a mixture-of-experts MoE approach to dataset distillation DD aimed at mitigating cross-architecture performance degradation. Traditional DD methods struggle when the distilled dataset is applied to architectures different from those used in the distillation process. This work tackles this limitation by involving multiple expert models, each responsible for distilling a distinct subset of the data, to enhance diversity within the distilled dataset. A distance correlation minimization strategy encourages experts to learn distinct representations, and a mixup-based fusion strategy further improves the generalizability of the distilled dataset. Experimental results show significant cross-architecture performance improvements, especially in low-data regimes.

**Strengths:**

Innovative Framework
The use of multiple expert models in dataset distillation is a novel approach that addresses the prevalent issue of cross- architecture performance degradation.
Comprehensive Experimental Validation
The paper provides a thorough set of experiments across different architectures and datasets, demonstrating the effectiveness of the proposed method.
Clear Methodology
The methodology is well-documented and includes ablation studies to justify the inclusion of distance correlation minimization and mixup-based fusion.
Improved Performance
The multi-expert framework consistently shows better cross-architecture performance than single-expert baselines, especially in low-data settings.

**Weaknesses:**

Theoretical Justification
While the experimental results are compelling, the paper could benefit from a more in-depth theoretical analysis of why the multi-expert approach performs better in cross-architecture scenarios.
Potential Overfitting on Small Datasets
The method may require further validation on very large datasets or real-world applications to confirm its scalability.
Limited Discussion on Limitations
The paper does not sufficiently address potential downsides or scenarios where the multi-expert approach may not yield improvements.

**Questions:**

Could the authors elaborate on how the specific values for mixup parameters (e.g., Beta distribution parameter) were chosen and whether these values impact performance significantly?
Has the method been tested on tasks beyond classification, such as object detection, to verify its generalizability?
Is there a limit to the number of experts that can be effectively used before diminishing returns set in?

---

### Note · Authors · 2024-11-25

I have read and agree with the venue's withdrawal policy on behalf of myself and my co-authors.